# Charismatic Embeddedness: A Cultural Starting Mechanism Generating Relational Goods in an Interreligious Field. An Analysis from Algeria

**M. Licia Paglione** * and Marco Luppi *

Dipartimento di Scienze Sociali e Politiche, Economia e Management, Istituto Universitario Sophia, 50064 Figline e Incisa Valdarno, Italy
* Correspondence: licia.paglione@sophiauniversity.org (M.L.P.); marco.luppi@sophiauniversity.org (M.L.)

**Abstract:** This article, entering into the debate on the influence of cultural factors on social action, highlights how a charismatic inspiration, as part of religious culture, could represent a relevant element in social phenomena. In particular, this article proposes an analysis of the role of a specific charismatic inspiration, in relation to the spirituality of the Focolare Movement (FM), in the interreligious field thanks to the "embeddedness" of the social action of its members of different religions in a specific charismatic culture. The analysis aims to understand whether and how this mechanism works by observing a specific Catholic–Muslim phenomenon developed in Algeria since 1966, using an interdisciplinary perspective between sociology and history and the case-study strategy, discovering that what we define as "charismatic embeddedness" could work as a "starting mechanism" generating "interreligious relational goods".

**Keywords:** cultural embeddedness; relational good; interreligious dialogue; gift; Algeria

## 1. Introduction

The relevance of the cultural dimension, and—within it—of religion, in conditioning social action and the forms of social relationship is a classic theme in Western sociological tradition. We need only think of the well-known work of the sociologist Max Weber ([1905] 2011) on the influence of the Protestant ethics on the development of capitalism. Along these lines, a further well-known concept in sociology is linked to the development of the Polanyian "embeddedness" (Polanyi [1944] 1974, p. 61). In Karl Polanyi's view, embeddedness describes the fact that social action is never exclusively individual and free because it is immersed within a network of social relations and in specific cultures that affect the social actors. The relevance of the embeddedness—in particular, a cultural one—of social action can be recalled today through an analysis of various contemporary phenomena. In this article we use this concept, originated in the economic area, moving it into the field of interreligious studies, in order to investigate a historical event in which religion and in particular a charismatic inspiration—considered part of the cultural structure—is capable to influence social action. We call this influence *charismatic embeddedness*, and can be observed by its specific effects at the level of interreligious relationships. Before proceeding, let us better clarify the conceptual meaning of "embeddedness" and, more specifically, what we mean by *charismatic embeddedness*.

## 2. *Charismatic Embeddedness*: When the Social Action Is *Immersed* in Religious Culture

"Embeddedness" is a concept, born in Western economic sociological tradition, used to describe the influence of the relational and cultural dimension on social action, and especially, in the seminal definition, economic action. The notion may be traced back to sociologists and anthropologists Karl Polanyi and Clifford Geertz, and it was revitalized by Mark Granovetter (1985); since then, it has become a central concept in the new economic

sociology. Granovetter's influential article focuses on the embeddedness of economic action "in networks of interpersonal relations" (Granovetter 1985, p. 504). This scholar used the expression "structural embeddedness" (Granovetter 1985, p. 486) to indicate that not only do personal relations (the "relational" aspect of embeddedness) matter, but also the "the structure of the overall network of relations" (Granovetter 1990, pp. 98–99). In Zukin and Dimaggio's (1990) wider conception, four kinds of embeddedness of economic action are identified: cognitive, cultural, structural, and political embeddedness. Cognitive embeddedness refers to "the ways in which the structured regularities of mental processes limit the exercise of economic reasoning". This notion calls attention to "the limited ability of both human and corporate actors" (Zukin and Dimaggio 1990, pp. 15–16) to employ the kind of rationality required by neoclassical economics. Cultural embeddedness refers to "the role of shared collective understandings in shaping economic strategies and goals" (Zukin and Dimaggio 1990, p. 17). Structural embeddedness is defined, following Granovetter, as "the contextualization of economic exchange in patterns of ongoing interpersonal relations" (Zukin and Dimaggio 1990, p. 18). Finally, by political embeddedness, the scholars mean "the manner in which economic institutions and decisions are shaped by a struggle for power that involves economic actors and non market institutions", such as the legal framework of the state (Zukin and Dimaggio 1990, p. 20). In this article, we propose to move this concept into the interreligious field and to consider a specific form of cultural embeddedness, linked to the religious dimension and, within it, to specific charismatic inspirations,[1] like those emerging from new spiritual movements spread within the Catholic Church in the past century. Thus, we propose to define this form of embeddedness as *charismatic embeddedness*.

### 3. Is *Charismatic Embeddedness* Generating *Relational Good* in the Interreligious Field? Research Question, Theoretical Perspective and Methodology

Using this concept, the article presents a research aiming to understand whether and how the religious culture, and into this, a particular charismatic inspiration, could embed social action and influence the quality of relationships in an interreligious field. To understand this, the research adopts a specific sociological framework useful to analyze relational dimension: the recent *Paradigm of Gift* (Caillé 1998, 2008; Godbout 2002, 2008). The origin of this perspective is rooted in anthropological observations of a gift, a form of exchange (Malinowski [1922] 1989; Mauss [1925] 2002), universally widespread in ancient and traditional societies (Māori, Polynesian, Melanesian...), constituted by three actions linked together in a cycle (giving, receiving, and giving in return) and oriented to a specific sociological function: building relationships and alliances among groups. According to anthropological observations, this function was possible because the concrete objects circulating as gifts had no primary economic or material value, but a symbolic one: they contained a spiritual power (*hau* in Māori language), a "driving force" which drew out from within the receiver a sort of obligation, a free obligation, to reciprocate. This obligation was free because the receiver could direct the counter-gift to others different from the first giver; the receiver was not conditioned in terms of the way, deadline, or typology of the counter-gift. Furthermore, even if the gift seemed to be totally gratuitous, it revealed deep down an interest, a particular interest in building bonds. In this paradoxical frame, the gift exchange spread by reciprocal actions pushed by a combination of "mixed motivations" (obligation/freedom, interest/gratuitousness) (Caillé 1998, p. 43), and in this way created a "structure of reciprocity" (Pulcini 2005, p. 198), or, in other words, relationships between "symmetrical groups" (Polanyi [1957] 1978, p. 306), i.e., between equals. According to recent scholars, in modern and contemporary societies, the gift carries on its function as "principal operator in building social bonds" (Caillé 1998, p. 83) thanks to its specific value, different from the exchange value and the utility value, called "value of bonding" (Caillé 1998, p. 9). This value expresses, in other words, the quantity of "gratuitousness" contained in the gift, or better yet, in the *animus*, i.e., the motivation, of the gift-giver and the receiver. And, according to contemporary scholars, this motivational element

qualifies the relationships. If the gratuitousness is the principal motivation between the four defined by Mauss in the social action—in the starting moment and in every moment of the gift cycle—an unconditional logic-i.e., the ability to move toward others without an exact guarantee of receiving something in return, using the courage derived from the philosophy to "leap into the unknown" (Caillé 1998, p. 122)-prevails, thus qualifying the reciprocity as "unconditional" and the relationships as "properly human" (Godbout 2002, p. 175). If, vice versa, in the social action, other motivations, such as obligation and interest, prevail, a conditional logic and "relationships of power" or "instrumental relationships" emerge. As the socio-anthropological observation shows, the gift-exchange produces potentially ambiguous relational effects. As Caillé (2008, pp. 32–33) explains, the analysis of the vocabulary reveals that the word *gift* in English means something gratifying for the participants in the exchange, but in German means *poison*, something capable of killing (according to the *dosys*, i.e., the combination of different motivations. The "properly human relationships" that the gift creates when gratuitousness and unconditionality prevail are relationships in which the individuals are able to express themselves, and be recognize as a whole. Simultaneously, these relationships are based on the consciousness of the incompleteness of the individual, and therefore the need of others as a "constitutive dimension of one's own identity" (Pulcini 2005, p. 194). For this element, the properly human relationships are very similar to a type of relationship, based on unconditional reciprocity that are inclusive and, at same time, capable of saving different identities, which, in the scientific debate on human flourishing, is known as "relational goods", which we will also use herein (Gui 1987; Donati 1986, 2019; Ulhaner 1989; Nussbaum 1986; Bruni 2004; Donati and Solci 2011; Paglione 2018, 2023). Following the *Paradigm of Gift*, it is possible to identify "lenses" (Table 1) to observe different typologies of social relationships, according to different motivations and logics of the social action.

**Table 1.** "Lens" from *Paradigm of Gift*.

| "Lens" from *Paradigm of Gift* | | |
|---|---|---|
| Social action | Motivations | obligation, freedom, interest, gratuitousness |
| | Logic | unconditional conditionality |
| Types of social relationships | | - instrumental relationships; <br> - relationships of power (or asymmetric relationships); <br> - properly human relationships or *relational goods*. |

Using the concept of *charismatic embeddedness* and the theoretical perspective described, an empirical research in the interreligious field was conducted, aiming to understand whether and how specific charismatic-embedded social actions could affect the quality of interreligious relationships. Using the case-study strategy, the research focused on the historical evolution of a specific interreligious phenomenon recently rediscovered (Callebaut 2021; Catalano 2022; Driessen 2023): the experience of a Catholic–Muslim dialogue developed since the 1960s in Algeria by the community of the Focolare Movement (FM), an innovative charismatic movement in the Catholic Church born in Trento (Italy) by Chiara Lubich in 1943 and characterized by a specific charismatic vision, called Spirituality of Unity[2]. Observing this phenomenon, the research used a qualitative design and tools such as content analysis of personal and public documents and semi-structured interviews with a first-hand witness of the phenomenon. The documents, offered by the International Center for Interreligious Dialogue of the FM in Grottaferrata (Rome, Italy), covers the period from its origins in the 1960s to the mid-2000s, and consists of life stories, public speeches, and edited interviews. The semi-structured interviews were realized with the coordinator of this center in different moments in order to clarify some central topics emerging from the content analysis of the documents.

## 4. The Catholic–Muslim Dialogue Promoted by the Focolare Movement in Algeria: A Case of *Charismatic Embeddedness* in an Interreligious Field and Its Relational Effects

Since 1966, some members of the FM moved to Algeria, in Tlemcen, a territory with a predominantly Muslim population, to take over the management of a former Benedictine monastery, which took the name "Dar el salam", to be transformed into a center for meetings and spirituality. Although of Catholic origin, the FM could be considered an interreligious movement because it involves people with other religious denominations and members of various other religions (Buddhism, Judaism, Islam, traditional African religions, Hinduism, Sikhism). This interreligious aspiration is in line with the charismatic vision, expressed by the central idea in the Spirituality of Unity: "making humanity one family". What happened in Algeria from the 1960s onwards—and which continues to exist today in the form of a community experience—is one of the many events animated by this particular inspiration and interreligious sensitivity, which can be transformed into good practices that have an impact on social reality.

### 4.1. The Algerian Context: A History of Conflict and Dialogue

In order to contextualize the starting moment of this phenomenon in Tlemcen in the 1960s, it is necessary to bear in mind some peculiar features of Algeria's national and regional history, which offer important information on the territory and the people. The country, in fact, is best known for national events related to the contemporary era, when the Regency of Algiers, led in a similar manner to the other two districts (Beylik) of Oran and Constantine by an Ottoman governor, entered the orbit of French intervention from 1830 onwards. Algerian resistance grew its fundamental nuclei in the east, Constantine, under the command of Ahmed Bey, but especially in the west with Emir 'Abd al-Qādir who, starting in the province of Oran and seeking the agreement of the local Berber populations, attempted to seize Central and Western Algeria by acting as a catalyst for a liberation jihad. After the conclusion of the First Franco–Moroccan War in 1844 and the agreements imposed on the Sultan of Morocco, who had sought to support the uprising in an anti-French capacity, 1848 marked the establishment of the conditions for incorporating the entire Algerian territory among the French departments (Julien 1970, pp. 273–335; Naylor 2009, pp. 57–140; Mc Dougall 2017, pp. 9–85). Algeria, a French "creation" (Pervillé 2019, p. 51)[3]—which, like several other contexts in colonial times, had been diplomatically designed in an arbitrary manner and without any particular reference to previous history— was at that time beginning a parenthesis of instability and conflict. The Algerian territory, which had become a colony since 1883 and was considered to be on par with the mother country[4], was subject to substantial emigration, which, by the mid-twentieth century, saw more than a million white or assimilated settlers as a result of mixed marriages or due to naturalization processes (Breil 1960, p. 219)[5]. This community, the majority of which was entrenched in nationalist ideological positions, marked by Eurocentric impulses and racially motivated evaluations, determined a form of coexistence based on exclusion and tension, well represented by the structure of the military barracks, the one that the Martinique Frantz Fanon described in *The Damned of the Earth* as the symbolic element of the French overseas government[6]. Legislators and rulers, within the complex colonial context, as Ofrath's work has recently shown, did not so much use the tools of dialogue but rather confirmed a subjugation, incentivizing settler privileges, the exclusion of the Muslim population, and in general, a concept of citizenship hostile to social and religious pluralism (Ofrath 2023). Some authors also use the definition of "colonial trauma" for the Algerian context (Lazali 2021; Benrabah 2013). The tight control, the police regime, and the reality of discrimination therefore did not facilitate an experience of coexistence and citizenship, but rather the emergence of an indigenous counter-nationalism, which set as its fundamental objective the process of enfranchisement from the rulers. This culminated in a war of independence (1954–1962) carried out by the Algerian FLN (National Liberation Front) against the French government and its army, in a period marked by atrocities and violence (terrorist attacks, round-ups, and urban guerrilla phenomena), carried out by both sides in a systematic

nature rarely seen before[7]. The intervention of De Gaulle, who was appointed Prime Minister in June 1958, led firmly the political process, even when it came to repressing the extremist attempts of the Franco–Algerian factions (the structuring of the OAS, the Salan coup d'état on 1961), which did not share the dialogue initiative opened by the Marshal with the native components and which fed a narrative linked to the practice of grandeur and imperialism necessary for France's prestige. The process experienced its final sanction on 1–3 July 1962, when the self-determination referendum and the proclamation of Algerian independence took place[8]. If the contemporary era has predominantly experienced marked conflict, as result from the French imperialist season indigenous claims that Ottoman rule was incapable of managing or guiding toward projects of self-determination—a history in which the religious factor failed to mitigate the dramatic course of events—, where can one trace useful elements to explain the emergence of conditions for a complex but possible interreligious dialogue such as that which took place at the Tlemcen community?

Algeria is historically a territory that has hosted the development of monotheistic religious communities, belonging to the Abrahamic family. Algeria, at the time known as Numidia and was a part of the vast dominions of the Roman Empire (Naylor 2009, pp. 35–56), had contributed significantly to the development of Christianity. Augustine was a native of Tagaste and worked as a bishop for over forty years in Hippo, an area now known as Annaba, along the northeastern coastal strip near the border with Tunisia. The Christian community in the area, the fruit of the first evangelization journeys, went through the controversies related to the first dogmas of the faith, with Augustine strongly polemizing against Donatism and Pelagianism, in order to defend the truthfulness of a doctrine that was still being consolidated. The Byzantine period, which began under Justinian in 534 and lasted for over a century, also represented a phase of contamination, traces of which remain through the architectural tradition of important basilicas and decorative art with the schools of mosaics, ceramics, and glass. What did not shine within a territory with solid Christian roots was the administrative project led by Constantinople, which seemed far removed from the needs of the local communities, for whom a substantial commonality of faith did not justify what they considered an example of misrule (Diehl [1896] 2018; Stevens and Conant 2016). The Arab invasion of the 7th–8th centuries constituted a real watershed event for all of North Africa. Beginning with the Rustamid dynasty, which founded its kingdom among the Berber tribes around 779 near the city of Tahert, there was an alternation of Muslim dynasties in Algerian territory, from the Umayyads to the Marinids, which developed a prevailing culture with the Arabic language and Islam an element of its identity[9]. There remained, in this period as in the long interlude of Ottoman rule, a minority presence of other religious communities, especially Jewish and Christian. Subsequently, the coexistence between religious cultures, profoundly conditioned by historical–political events, resulted in a new increase during the era of the French conquest, where more than one million non-Muslims were mentioned during the 1960 census, the majority of whom were Catholics (Breil 1960, p. 219). A new situation of tension arose following independence, with the dramatic upsurge associated with the Algerian civil war (1991–2002)[10] when Christian communities were also subjected to attacks by armed extremist groups. Among the events that shook public opinion, both locally and especially abroad—especially in France—was the execution of the Bishop of Oran, Mgr. Claverie, on 1 August 1996, who was actively engaged in Islamic–Christian dialogue, so much so as to be known as the "bishop of the Muslims". The bishop was convinced of the possibility of inter-religious coexistence based on concrete praxis and after the painful affair of the Trappist monks of Tibhirine, who were kidnapped and slaughtered between March and May 1996 by an armed group with a still rather dubious identity, which seemed to operate among the acronyms that made up the insurgency. The monks' testimony was that of not abandoning the monastery and the style of their presence, based on inter-religious dialogue, even in the face of the concrete possibility of losing their lives in the name of such ideality[11].

*4.2. The (Hi)story of a Catholic–Muslim Interreligious Phenomenon in Algeria Embedded in a Charismatic Culture*

A history that has thus known alternating events, made up of divisions and conflicts, but with important personalities and experiences linked to inter-religious sensitivity and the practice of dialogue, may represent a particularly favorable terrain for welcoming the particular experience analyzed here. Tlemcen was a small town close to the border with Morocco, which today has a population of around 140,000 inhabitants, close to Oran, one of the prestigious outlets toward the Mediterranean. The city, capital of the eponymous kingdom founded during the final stages of the Almohadi dynasty, was ruled by sultans of the Zayyanid dynasty until the Ottoman conquest in 1554. Strategically located, it was one of the nerve centers of the trans-Saharan trade routes. Tlemcen in the 1960s was a predominantly Muslim locality, with a dwindling Christian presence, characterized by a popular religiosity open to foreigners and non-natives, far from the strongly identitarian and anti-Western currents that would emerge later in the 1970s. In 1966, just a few years after national independence, some members of the Catholic FM tradition moved here. As Driessen (2023) remembers, in his recent publication, the event happens "at a time when the community [of FM] had not yet developed a clear vision or theology of interreligious dialogue and was not looking to expand into Muslim North Africa"[12]. For some time, the Focolare, interested in pursuing the challenge of concretizing the idea of humanity as a single family, uniting members belonging also to different religions, had received an invitation to interweave relations with the Islamic world, in the conviction that there were various points in common linked to a form of community religiosity. The Algerian one was therefore the first opportunity, linked to the management of the Benedictine monastery. The community of FM sent three members to take up residence in the monastery and establish a Focolare, a home of celibate laymen (or women), which forms the central core of the Focolare community. The three men renamed the monastery Dar-Es-Salam and dedicated it as a center intended to create friendship for and with the local community. The small group knew very little about Islam or Algeria. Archbishop Henri Teissier, who became bishop of nearby Oran in 1972, worked closely with the community to help them articulate their presence in Tlemcen and, later, in Oran and Algiers, in harmony with the Algerian Church's developing vision of its non-proselytizing presence in Algeria, a vision which the FM members eagerly embraced. These first three people, Salvatore Strippoli, Ulisse Caglioni, and Pierre Le Vaslot, set themselves the first task of getting to know and interact with the local population. According to them, their involvement can be divided into three periods (Callebaut 2021, pp. 161–62). A first moment, between 1968 and 1969, involved some Algerian high school students, who discovered through interaction with their teacher (Pierre Le Vaslot) a life proposal that fascinated them to such an extent that they became personally involved, giving birth to a first interreligious spiritual group. A second period, of relative suspicion toward Europeans and therefore of greater silence and detachment, was marked by a certain dispersion of that first group, with some becoming more involved in study, and others accepting a stronger political involvement. In the meantime, the permanent members of the center took the opportunity to devote themselves mainly to learning the Islamic language and culture. A third period, relating to the 1980s, took place from the moment when some of the members of the initial group redeemed the relationships previously built, beginning to create a small community, a collective reality that adhered, while remaining Muslim, to the FM's spiritual proposal, recognizing the common faith in one God. Subsequently, in the 1990s, thanks to the establishment of a more stable group and through the daily experience developed in Tlemcen, interreligious relations were enriched, generating innovations, through, for example, the realization of international meetings of Christian–Islamic dialogue, attended by people of the Islamic religion from the five continents. From those years until today, opportunities for collaboration continue to be generated at many levels. In entering a predominantly Muslim context for the first time, those first three Catholics brought with them a cultural baggage steeped in the particular charismatic vision of the Spirituality of unity rooted in the religious movement to which

they belonged. A spirituality, in sociological terms, can be understood as a particular "way of life" (Séguy 1998, p. 149). The spirituality of unity can thus be considered a particular "way of life" emerged in the Catholic Church as an innovative force of charismatic nature, from the 1940s onwards, and was rooted in a specific culture that, for the purposes of this analysis, is useful to recall in order to grasp its influence on the social actions of its members and sympathizers in that new context. The core elements of this culture could be described from the systematization provided by the MF members themselves. A first element is the particular objective: unity, the creation of a more united humanity. The "way of life" proposed to realize this objective emphasizes the communitarian dimension and suggests twelve guidelines, called "cornerstones", strongly guiding the actions of the members of the FM (Lubich 2002, p. 14). Among these points, five—linked to each other—seem particularly relevant. Beginning with a particular understanding of "God as love" (point 1), i.e., the perception of divinity in paternal terms, others are seen as brothers to be loved. Hence, the idea of acting toward them, implementing "love of neighbor" (point 2), giving rise to a form of exchange defined as "mutual love" (point 3), which structures bonds of "unity" (point 4), fraternal relations, and presupposes a certain capacity to accept the risk of non-reciprocity, which in the terms of the Spirituality of Unity is expressed by the image of "Jesus forsaken" (point 5), a God who for love of man was willing to die. In summary, Lubich herself during the 1999 WCRP Assembly in Amman presented this way of life using these words:

> "But in what does this art of loving consist? First of all to love everyone indiscriminately, even Buddhists, even Muslims, even atheists; to love everyone indiscriminately: blacks, whites, men, women, small, large, Germans, Italians, Americans, South Americans. This is the first essential point. Second: love first, without expecting to be loved. Try to do this during the day with everyone you meet: at home, with the family, with the husband, with the wife, with the children, in the office, at school, in parliament; try to love first and see what comes out. Out comes the Christian revolution! Then again it consists not in the words: 'I love you', but in doing, in serving, which in two words is said: 'make yourself one' with the other, understand the other; if he suffers, suffer with him; if he enjoys, enjoy with him; if he thinks something, think with him; make yourself one with him. That is what St Paul asked, when he said to make yourself one with everyone. It is a toil, but it bears immense fruit. To make oneself one, in these two simple words lies the secret of that dialogue that can generate unity. Making oneself one that is not a tactic or an external way of doing things, it is not just an attitude of benevolence, openness, respect, or absence of judgement; it is not just bringing the poor man a little package, etc. It is certainly all that, but with something more"[13].

The Focolare charismatic culture, centered on this type of love, can be described through these guiding points that influence social action by directing it to generate a relational effect, i.e., "unity" or so-called "fraternal" relations which we could consider here equivalent to "properly human relationships" or "relational goods", types of relations emerging from the particular form of exchange that is the gift (Paglione 2018). The typology of social action above described appears very similar, in sociological terms, to the social action in the gift, with gratuitousness and unconditionality as principal motivation and logic and the creation of alliance as sociological function.

### 4.3. Social Actions Embedded in a Charismatic Culture and Its Relational Effects: Main Results

Adopting the *Paradigm of Gift*, the analysis focused on social action—observed according to two dimensions: motivations and logic—and on the quality of relationships emerging, aiming to highlight influence of FM charismatic culture. At the social action level, the first dimension, i.e., the motivation prompting social actors to act, can already be grasped in the first major action observable in the case analyzed: the transfer of FM members to Tlemcen. This act can be interpreted as a response to what their religious

culture demanded, an act in this sense, on the one hand, obliged and, on the other hand, instrumentally interested in introducing Christianity into the Muslim world. The obligation in the gift, however, left great margins of freedom, because the FM members responded by being there, creatively, imagining ways to enter into relationships. Moreover, the interest in this was not unilaterally pursued and did not translate into a desire to convert people of the Islamic religion to Christianity. This is clear when observing the meaning given to moving to Tlemcen by one of first FM members: for Ulysse Caglioni, a mechanic, the motivation was "the desire to love" and "to give continuity to relationships with them" (Cocchiaro 2006, p. 37). Testimonies confirm this motivation. They describe Ulysse as a man of few words, who quickly became integrated and appreciated by the Algerian people, made up of simple, hospitable people, with little intellectualized religiosity, who—according to Cardinal Duval of Algiers—needed fraternal love and lived in unity to understand Christianity (Cocchiaro 2006, p. 57). A witness to this was Didier Lucas, who stayed in the 1970s for several years in Tlemcen and returned after Ulysses' death. Remembering this figure, Lucas said:

> With Ulysses one did not make many speeches, life was enough. We understood each other immediately, there were no problems or difficulties. One thing that always struck me was that for his brother he gave everything and forgot the rest. Even if there were other things to be done, which I considered important, for example a meeting to prepare, he always postponed everything: he was concerned about making his brother happy (Cocchiaro 2006, p. 81).

If it was therefore an interest that moved Ulysses, it was an interest capable of "leaping into the unknown" (Caillé 1998, p. 122), accepting the risk of gratuitousness. Still, with respect to Ulysse, a Muslim woman, recalling tragic periods in Algerian history in the 1990s, wrote of him:

> Ulysses' love for God and the love he showed for our land was much stronger than fear and violence. To support us, he played down what was happening around us every day and he did so above all through his behavior (Cocchiaro 2006, p. 94).

Therefore, there was an interest conditioned by the religious culture of origin on the part of the FM members in moving to Tlemcen, but it was "mixed", in the *Paradigm of gift* terms, with a capacity of gratuitousness and a prevalently unconditional logic, such as the actions creating "properly human bonds" or "relational goods". In this sense, the words of a person of the Islamic religion relating to those years are interesting, concerning some qualities of the interreligious relations generated:

> We would often visit these young Christians; we would learn songs, do some chores, but the best moments were when we exchanged impressions and experiences. Then our relationships became deep, each was ready to listen and welcome the other. This led us to such an extraordinary group life that we became engrossed; each one was sure of being able to count on the others and the circle of friends grew wider each time (Cocchiaro 2006, p. 49).

The reciprocal exchange that underpinned such relationships, beyond the specific content, was animated by a strong "desire for bonding" in itself, as gift theorists would say, which nurtured the genesis of not instrumental or power-based relationships, but rather properly human ones: inclusive bonds wherein the specific identities, including religious identities, of the people involved counted and the diversities were preserved. Taking a leap in time and arriving more than 40 years after that first Catholic FM group moved to Algeria, we can better observe this last aspect. This can be seen, for example, in the account given years later by a Muslim woman who came into contact with the FM:

> Thanks to this movement I am moving forward on my journey of faith as a Muslim. [...] I started to read the Koran with my heart and a vision full of love. I think that if I had read the Koran before I met the way of Clare I would have

understood and interpreted it differently [...] my religion has now become life and the commitment to live it in my daily life helps me to be a better Muslim. We do not experience a mixture of religions but rather these encounters strengthen each in his own religion (Catalano 2010, p. 75).

What emerges, therefore, are religious identities that are well defined, but unconditionally open to welcoming the other and his diversity. Very significant, in this sense, is, for example, the case of a mixed couple formed by an observant Muslim and a practicing Catholic, married and close to the FM spirituality. Their bond is founded on and nourished by the diversity of which each is the bearer. To nurture their bond, it is essential to mutually cherish each other's religious identity. Speaking of prayer, for example, Barbara, the wife, shows how her husband, while remaining Muslim, supports her in being a good Christian:

There had been an attempt to each learn a prayer from the other, but then it seemed more consistent for each to remain themselves. Bahaman proved to be very open and mature, offering several times to accompany me to mass[14].

The preservation of identities in this interreligious relationship goes together with another effect: in each actor or group, the consciousness and vital experience of a common belonging become more marked, allowing a form of mutual integration, while respecting diversity. This effect emerged, for example, during the summer meetings promoted by the FM with Muslims starting in the 1980s in Algeria, when a sentence from the Gospel was identified every day to be lived out, as a guide for the day. This led the Muslims present to propose that a similar phrase also be identified in their scriptures to be shared with Christians. The commitment to live these phrases and share what they had lived had the effect of creating a living community, a sui generis reality, not only of Christians, but of Christians and Muslims, who experienced a form of mutual religious integration and common belonging to the same spiritual family.

This is shown by this excerpt taken from the account of a Muslim participant in the interreligious experience analyzed:

I own a printing house and some time ago I started to print some publications for the Focolare. I had many hesitations, also because I am Muslim from birth. Knowing more about another religion was difficult for me, but as time went on I felt that these people work for humanity, to promote brotherhood, peace. Then I started reading the Word of Life, which every month brings Clare's message and many experiences on how to build a united world. I came to the conclusion that all that is in our great book, the Koran, which is the path indicated by our prophet Muhammad, is the same path that Muslims and Christians have in common, and together we must take them forward" (Catalano 2010, pp. 74–75).

Similar elements emerge in Driessen (2023). According to this researcher, "various members expressed unease at describing their activity in the Focolare community as a form of interreligious dialogue, as if that implied their primary action and sense was to be in dialogue, to increase their knowledge about Christianity, or to explain Islam to others. The theological framework of the Algerian community, in this sense, is quite loose and unimposing. The lived experience is primarily one of building community, of sharing with and serving one another" (Driessen 2023, p. 142). In this sense, Driessen remembers the words of a longtime Muslim member in Oran:

This is not dialogue. This is living religiously together. We are not trying to figure Christians out. We are within the community. We are already in the family, sharing everything, and that is not a problem. Dialogue is surpassed (Driessen 2023, pp. 142–43).

A Christian member of the community in Algeria put it this way:

We forget that they are Muslims and vice versa. We are walking together on a path towards the truth, in confidence, trusting in the experience of love, of the

Muslims themselves, that God is there, that we are all one, that God is bigger than all that (Driessen 2023, pp. 142–43).

What had slowly been generated in Algeria was a group deeply rooted in the FM, but Muslim and engaged in a process of creating interreligious relationships. The type of relationships between members of different religions, at different levels, seems to be underpinned by a mutual exchange rooted in the recognition that religious diversity is a fundamental value, nurturing the bond. From the life story of a participant at an Islamic–Christian conference (Verona on 2008), coordinated by FM, with around 700 Muslim and Christian attendees, emerges a testimony on the importance of this element:

> Together—it is a common voice on both sides—we discover the beauty of each as a mutual gift, the commitment to live to contribute to achieving a world united in fraternity. These are moments that we live in the spirit of universal brotherhood: we experience a great joy in coming together, in sharing experiences, in discovering in an ever greater way that realizing a world of brothers and sisters is not a utopia, but a dream that can be realized, because we feel that among us it is already a reality (Catalano 2010, pp. 74–75).

The relational effects linked to this type of charismatically embedded action thus seem to coincide with the genesis of relationships nourished mainly by unconditional reciprocity, in which the people involved and the relationships between them are considered values in themselves and not instruments to obtain something else, very similar to "properly human relationships" or "relational goods" in an interreligious field.

## 5. *Charismatic Embeddedness* as a Cultural Starting Mechanism Generating *Interreligious Relational Goods*

From the results of our analysis, therefore, it is possible to grasp how, in the phenomenon observed, the charismatic embeddedness represents a cultural structure influencing the social action and improving the genesis of relationships between equals which are inclusive and, at the same time, respectful of different identities in an interreligious field. Indeed, it is possible say that in the interreligious Algerian context, the specific *charismatic embeddedness* linked to the culture of FM spirituality seems to work as a *cultural starting mechanism* creating a specific type of social action—where gratuitousness as motivations of the action prevails—and a specific type of interreligious relationship, that we could call *interreligious relational goods*. This mechanism, in the case analyzed, could be seen as the initial *driving force* behind the development of an experience that Fontaine (2016) could identify, in the particular trajectory experienced by Christianity in Algerian history, as a case of interreligious encounters historically capable of "making a difference". According to this scholar:

> While Christianity in Algeria was long used as a means to justify the colonial regime, various forms of settler privilege, and even the use of torture and extreme violence, it also became one important tool through which those practices could be challenged. (...) it was the individual believers who made the difference; in some cases, their choices had a global impact (Fontaine 2016, p. 224).

A context marked by the absolute prevalence of Islamic sensibilities and a Christian presence associated, in most cases, with particularly negative and divisive phenomena (colonization, prevarication, exploitation), had in itself all the elements to reject any attempt at community building under the banner of peaceful interreligious dialogue. What contributed to the introduction of a change in perspectives was the particular "way of life" and individual action, embedded in a particular culture in radical countertendency, capable of going beyond a path of decolonization, because in a certain way, they had preceded and contradicted it, showing concrete examples (Msgr. Claverie, the monks of Tibhirine) of a Christianity living under the banner of intercultural and inter-religious coexistence, ready to give its life concretely and unconditionally, that is, to make what Caillè would call a "leap into the unknown" for the Algerian well-being. The experience carried out by

Muslims and Christians who are charismatically embedded in the culture of the Focolare Movement in Algeria—Tlemcen was mentioned, but similar dynamics could be found in Algiers and Oran—seems to go one step further. The case analyzed, although limited and circumstantial, shows that a way of life immersed in a religious culture that values and animates gratuitousness and unconditionality in the social action, pushes people to transform religion from an element of division, generating tragic conditions, as it is in some contexts, to a "privileged operator of sociality" (Caillé 1998, p. 43), once again drawing on—evident in the etymological Latin origin of the term *religo*—the deepest sense of religious experience.

**Author Contributions:** Conceptualization, M.L.P. and M.L.; methodology, M.L.P.; formal analysis, M.L.P.; investigation, M.L.P. and M.L.; data curation, M.L.P.; writing—original draft preparation, M.L.P. and M.L.; writing—review and editing, M.L.P. All authors have read and agreed to the published version of the manuscript.

**Funding:** The research didn't received funding.

**Institutional Review Board Statement:** Ethical review and approval were waived for this study due to the fact that the data is part of the public domain.

**Informed Consent Statement:** Not applicable.

**Data Availability Statement:** Most of the data for this article come from the Archive of the International Center for Interreligious Dialogue of Focolare Movement located in Grottaferrata (Rome, Italy), all other sources are indicated following the requiered citation guidelines and within the Reference list.

**Acknowledgments:** The authors acknowledge support given from International Center for Interreligious Dialogue of the Focolare Movement (Grottaferrata, Rome, Italy).

**Conflicts of Interest:** The authors declare no conflicts of interest.

## Notes

[1]    The term *charisma* is used here as a synonym for "spirituality" to indicate a particular contribution to understanding the religious message in an original key to which a particular cultural perspective and a particular "way of life" is linked (Séguy 1998).

[2]    The case was recently put forward by M. Driessen in his book *The Global Politics of Interreligious Dialogue: Religious Change, Citizenship, and Solidarity in the Middle East* (Driessen 2023) and already studied by Callebaut (2021); Catalano (2022).

[3]    Pervillé writes: "Algeria is a human creation, the product of history. Although physical geography has provided the framework within which to inscribe its parable, it is not enough to explain it. What has generated the entity that we now call by this name is the history of the men who have inhabited and conquered it. (…) The bulk of its borders exist only on maps, geographical and mental. Nothing, in short, made it necessary for a state, let alone a power, to arise from that stretch of Mediterranean coastline. If it did, it was largely due to the colonisation of France. Of which Algeria is a North African inheritance" (Pervillé 2019, p. 51).

[4]    On 12 November 1954, at the initial moment of the Algerian rebellion against the French colonizers, French Prime Minister Mendès-France declared during a session of the French National Assembly: "One does not compromise when it comes to defending the internal peace of the nation, the unity and integrity of the Republic. The Algerian departments are part of the French Republic. They have been French for a long time, and they are irrevocably French. (…) Between them and metropolitan France there can be no conceivable secession".

[5]    Among the assimilated, in all respects, French citizens were the so-called pieds-noirs, the Frenchmen of European descent originally from Algeria, who, depending on the political events in the country, were forced to migrate permanently to France.

[6]    The first edition of F. Fanon's text, *Les Damnés de la terre*, published in 1961 in Paris (Éditions Maspero) a few days after its author's death, soon became one of the reference texts of the Third Worldist struggle. On the reality of Algeria, in terms of repression and violence, see Thénault (2012).

[7]    On these topics, see: (Pervillé 2002; Naylor 2000, pp. 23–73; Mc Dougall 2017, op. cit., pp. 179–234; Ageron 1991, pp. 93–144; Stora 2004; Calchi Novati and Roggero 2018).

[8]    It should be noted, however, that Algeria considered 5 July to be its national liberation holiday, as it corresponded to the anniversary of the seizure of Algiers by French troops.

[9]    On the first, long phase of Arab domination, the VII–XVI centuries, see: Naylor 2009 op. cit., pp. 57–108; M. Brett, *The Arab conquest and the rise of Islam in North Africa*, in Fage (1979, pp. 490–555); (Hoyland 2014).

[10]   On Algerian civil war see: (Martinez and Entelis 2002; Eldridge 2018; Vince 2020).

11    On the experience of the monks of Tibhirine and mgr. Claverie see: (Kiser 2002; Georgeon et al. 2018; Pérennès 2000; Monge and Routhier 2018).

12    On this topic see: (Coda 2000; Tobler 2022).

13    From Speech of Chiara Lubich during WCRP Assembly, Amman 1999 (Document from the Focolare Center for Interreligious Dialogue).

14    From *Mille musulmani intorno al focolare*, p. 161 (Document from the Focolare Center for Interreligious Dialogue).

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
