# Peer review of "Charismatic Embeddedness: A Cultural Starting Mechanism Generating Relational Goods in an Interreligious Field. An Analysis from Algeria"

_religions, doi:10.3390/rel15010058_

Round 1

Reviewer 1 Report

Comments and Suggestions for Authors

Some remarks regarding the Algerian history:

157. Algeria comprised three Ottoman regencies (Beylik): Algiers, Constantine, and Oran.

160. I am not sure we can say that Emir 'Abd al-Qādir was a “ruler of the province of Oran”. After the fall of the Western Beylik (Oran), he led a revolt against the French occupation for quite two decades.

165. What do you mean by “diplomatically” in “Algeria […] had been  diplomatically designed”? It is rather bloodily and violently. Perhaps you mean “Designed on the map in closed rooms.”

188. June 1858, you mean 1958.

215. “the Rustamid dynasty of Persian origin.” ‘Abd al-Raḥmān ibn Rusm’s father was a Persian. He was chosen to avoid the tribal competition, but the state was totally Berber.

220-221. “Aa shari'a-led kingdom” was the norm in the Islamic world. So, it is not something specific. I don’t know what you mean by “assimilation”, because the Maghreb had its autonomy very early, and the states were founded mainly by the autochthon Berber population.  

233. It is difficult to claim bluntly that the Trappist monks of Tibhirine were kidnapped and slaughtered by the “Armed Islamic Group”. Even the identity of this group is still ambiguous, most probably a creation of the Army services.

244. Tlemcen has a population of around 100,000 inhabitants. I think more, 140,000.

246. The Zayyanid “tradition.” It is instead a dynasty.

252. The first time you use the abbreviation FM (Focolare Movement), explain it.

311-327. the paragraph is without a reference. 

Author Response

Thank you for the suggestions

We worked in the article directly.

We included all suggestions and tried to clarify better some parts, connections, and the intent of the historical description of the context, e.i. highlighting the ancient multireligious experience of the Algerian context.

Please see the attachment (new version of the article with changes in red color)

Reviewer 2 Report

Comments and Suggestions for Authors

The article's main idea is interesting, captivating and quite original; however, the article structure, methodology and coherence is very problematic. 

The essay starts with a sociological analysis based on western economic -social theories difficult to associate with that Algerian world lately analyzed in his inter-religious context. I don't see any cognitive association between M. Mauss' gifts theory with Lubich's spiritual and charismatic experience.

Can a spiritual gift be related with a material one; in specific for a western society that has always encouraged the association between gift and good (as a commodity)?

In other words, in which way this relational perspective (pag. 3) can be inherently and spiritually associated with the gratuity of the Gift in the Focolare movement?

The Algerian historical context is not well introduced also and the passage from contemporary to the ancient and late-antiquity era, problematic. I could suggest  to read more about the Berbers-Kharijites social and political system in early Islam as the facets of their religious system, but honestly, I am not able to consider the author's concrete aim in proposing this historical excursus. 

Finally, the chp. 4.3 on Ulisse Caglioni's interrelation with the Algerian-Muslim world seems again settled in the article without a concrete and previous explanation about the context: the Focolare's praxis and theological core, the Algerian's experience, the emerging inner issues.  It seems that the article is framed by the association of three parts that made many difficulties in being rationally and methodologically held together. 

Author Response

Thank you for your suggestions.

We worked in order to response/integrate/modify in the article directly.

Here our answers ( -->):

The article's main idea is interesting, captivating and quite original; however, the article structure, methodology and coherence is very problematic. 

The essay starts with a sociological analysis based on western economic -social theories difficult to associate with that Algerian world lately analyzed in his inter-religious context.

--> A aim of the article is conceptual also, in the sociological debate. So the analysis used a sociological concept, developed in a specific tradition of the discipline yes, but with an innovative application due the link with religious filed (charismatic embeddednes), and the socio-anthropological perspective (Paradigm of gift) is useful to interprete a specific phenomenon as case-study, observing it in relational dimensions, as the research question asks. In this way the intent is to offer a key to interprete the evolution of interrreligious relationaships in a specific context, but also propose a new application of a concept (embeddedness)

I don't see any cognitive association between M. Mauss' gifts theory with Lubich's spiritual and charismatic experience.

Can a spiritual gift be related with a material one; in specific for a western society that has always encouraged the association between gift and good (as a commodity)?

-->We modifyed the article, describing in better way the paradigm of gift, clarifying that that it is the teoretical perspective to analyse the case-stady. The Paradigm of gift is a socio-anthropological perspective the gift is material and spiritual, at the same time. This form of gift was typical in not occidental societies observed by anthopologist as Marcell Mauss, B. Malinoski…

The question for the scholars is wheter it is possible to try gift in contemporary societies…mixed with the commodities… and able to give them a another value, the “value of bonds”. The sociological function of the gift was and is creating alliances, human relationships, “relational goods”. We modified the article to try clarying better.

To search the presence of gift in contemporary societies is the challenge, and many scholars work in this line (Caillé, Godbout, Fistetti, Brunii). Our article is in this line also.

In other words, in which way this relational perspective (pag. 3) can be inherently and spiritually associated with the gratuity of the Gift in the Focolare movement?

-->We modifyed the text to clarify. The Paradigm of gift for us is the theorethical perspective to analyze empirically the case-study. In the gift dynamic the gratuitness is a motivation in social action essential to qualify the object-gift and the relationships created by gift-exchange as properly human or as “relational goods”. In FM’s culture gratuitness is called love.

The Algerian historical context is not well introduced also and the passage from contemporary to the ancient and late-antiquity era, problematic. I could suggest  to read more about the Berbers-Kharijites social and political system in early Islam as the facets of their religious system, but honestly, I am not able to consider the author's concrete aim in proposing this historical excursus. 

--> We tried to clarify better and show the connections. The intent of the historical description of the context was highlighting the ancient multireligious experience in the Algerian context.

Finally, the chp. 4.3 on Ulisse Caglioni's interrelation with the Algerian-Muslim world seems again settled in the article without a concrete and previous explanation about the context: the Focolare's praxis and theological core, the Algerian's experience, the emerging inner issues.  It seems that the article is framed by the association of three parts that made many difficulties in being rationally and methodologically held together. 

--> We connected better Ulysse Caglioni. We described in few words (because the article is not theological) the Focolare theological core in lines 308-326. And we tried to unify better the parts and clarify methodology.

Please see the attachment (new version of the article with changes in red color)

Round 2

Reviewer 2 Report

Comments and Suggestions for Authors

The revision has been made, the article can be published now.